# Anti-Tumor Effects of a Penetratin Peptide Targeting Transcription of E2F-1, 2 and 3a Is Enhanced When Used in Combination with Pemetrexed or Cisplatin

**DOI:** 10.3390/cancers13050972

**Published:** 2021-02-26

**Authors:** Gulam Mohmad Rather, Michael Anyanwu, Tamara Minko, Olga Garbuzenko, Zoltan Szekely, Joseph R. Bertino

**Affiliations:** 1Rutgers Cancer Institute of New Jersey, Rutgers, The State University of New Jersey, New Brunswick, NJ 08901, USA; gmr112@cinj.rutgers.edu (G.M.R.); mikeomega781@gmail.com (M.A.); minko@pharmacy.rutgers.edu (T.M.); zoltan@pharmacy.rutgers.edu (Z.S.); 2Department of Pharmaceutics, Ernest Mario School of Pharmacy, Rutgers, The State University of New Jersey, Piscataway, NJ 08554, USA; olgagar@pharmacy.rutgers.edu; 3Department of Chemistry and Chemical Biology, Rutgers, The State University of New Jersey, Piscataway, NJ 08854, USA; 4Department of Pharmacology and Medicine, Robert Wood Johnson Medical School, Rutgers, The State University of New Jersey, New Brunswick, NJ 08901, USA

**Keywords:** E2F, cisplatin, pemetrexed, thymidylate synthase, thymidine kinase, NSCLC

## Abstract

**Simple Summary:**

The E2F family of transcription factors are essential for cell proliferation, differentiation, and DNA repair. They are commonly overexpressed or dysregulated in cancer as a consequence of inactivation or mutations in the retinoblastoma protein. Therefore, one or more of the activating E2Fs (E2F-1, 2, and 3a) have been recognized as antitumor targets. The combination of a peptide targeting transcription of E2F-1, 2, and 3a, with cisplatin, and especially with pemetrexed, showed enhanced antitumor activity *in*-*vitro* and *in*-*vivo* and has promise for the treatment of patients with various tumors, and in particular, lung adenocarcinoma.

**Abstract:**

Background: We tested the antitumor effects of a modified E2F peptide substituting D-Arg for L-Arg, conjugated to penetratin (PEP) against solid tumor cell lines and the CCRF-leukemia cell line, alone and in combination with pemetrexed or with cisplatin. For *in*-*vivo* studies, the peptide was encapsulated in PEGylated liposomes (PL-PEP) to increase half-life and stability. Methods: Prostate cancer (DU145 and PC3), breast cancer (MCF7, MDA-MB-468, and 4T1), lymphoma (CCRF-CEM), and non-small cell lung cancer (NSCLC) cell lines (H2009, H441, H1975, and H2228) were treated with D-Arg PEP in combination with cisplatin or pemetrexed. Western blot analysis was performed on the NSCLC for E2F-1, pRb, thymidylate synthase, and thymidine kinase. The H2009 cell line was selected for an *in*-*vivo* study. Results: When the PEP was combined with cisplatin and tested against solid tumor cell lines and the CCRF-CEM leukemia cell line, there was a modest synergistic effect. A marked synergistic effect was seen when the combination of pemetrexed and the PEP was tested against the adenocarcinoma lung cancer cell lines. The addition of the PEP to pemetrexed enhanced the antitumor effects of pemetrexed in a xenograft of the H2009 in mice. Conclusions: The D-Arg PEP in combination with cisplatin caused synergistic cell kill against prostate, breast, lung cancers, and the CCRF-CEM cell line. Marked synergy resulted when the D-Arg PEP was used in combination with pemetrexed against the lung adenocarcinoma cell lines. A xenograft study using the PL-PEP in combination with pemetrexed showed enhanced anti-tumor effects compared to each drug alone.

## 1. Introduction

The E2F family members are critical to many cellular processes, including development, proliferation, DNA repair, and differentiation [1,2]. All family members recognize a canonical binding sequence (5′-TTTSSCGC-3′ (S=C or G)) in the promoter of their target genes, including enzymes required for DNA synthesis that include dihydrofolate reductase (DHFR), thymidine kinase (TK), thymidylate synthase (TS), DNA polymerase alpha, and the R2 subunit of ribonucleotide reductase (RR). Of the eight family members identified to date, E2F-1, -2, and -3a act as transcriptional activators of many genes, while E2Fs 3b, 4, 5, and 6 act as transcriptional repressors. E2F-7 and 8 are not as well understood; they lack activation domains but have anti-proliferative functions [3]. E2F-1 is unique among the activating E2Fs in that overexpression can cause apoptosis in some cell lines, while driving proliferation in other cell lines [4,5].

A central player in E2F regulation is the tumor suppressor retinoblastoma protein (pRB) and family members. When under-phosphorylated, RB binds to the activating E2Fs at promoter regions and inhibits their transcriptional activity [6,7,8,9]. In response to signals favoring cell cycle progression, RB is phosphorylated by CDK4, 6, and 2, and E2F is freed from its repressor complex to activate the transcription of its target genes. Loss or mutation of the RB gene has been identified in many human malignancies [10] and leads to an increase of free E2F and cell cycle dysregulation [6]. A second less common mechanism of E2F overexpression is gene amplification; in this case, it is usually E2F-1 or E2F-3 that is amplified in human tumors [2]. Regardless of the mechanism underlying the overexpression of E2F, overexpression is associated with poor prognosis [11,12,13]. Indeed, it has been proposed that tumor cells that over express E2F-activating proteins may be “addicted” to this oncogene [14,15], making them particularly vulnerable to E2F downregulation. However, it should be noted that overexpression of E2F-1, may lead to apoptosis in cells with wild-type p53, while tumor cells that lack functional p53 due to mutation or loss are able to tolerate high levels of E2F-1.

Given the role of the E2Fs in cell cycle regulation and proliferation, there have been many attempts to downregulate E2F in order to inhibit tumor cell growth. For example, knockdown of E2F-1 by siRNA in castrate-resistant prostate cells lacking pRB decreased proliferation [16]. However, a growing body of evidence strongly indicates that it may be necessary to target all three activating E2Fs to cause marked inhibition of growth. Knockout of all three activating E2Fs was necessary to completely arrest cell growth in mouse embryonic fibroblasts (MEFs) and in mice [17].

Using phage display, we identified a novel peptide that inhibits transcription of E2F-1, 2, and 3a, and as a consequence their target genes, including TS, TK, and DHFR. When coupled to a penetratin to enhance uptake, the penetratin E2F peptide (PEP) was shown to be cytotoxic to a subset of tumors that overexpress E2F [18]. Moreover, when encapsulated in PEGylated liposomes (PL-PEP) to improve serum half-life and tumor targeting, the PL-PEP inhibited growth of mouse-borne xenografts of two human tumors, H-69, a small cell lung cancer, and DU145, a castrate resistant prostate tumor; both lack RB and harbor a mutant p53. The anti-tumor effect occurred without apparent host toxicity [18,19]. Recently, we have shown that substituting the L-Arg amino acid in the E2F peptide with D-Arg resulted in a peptide that is both more potent and more resistant to proteolysis [20]. Computational modeling studies confirmed that D-Arg PEP is more stable compared to L-Arg PEP and binds tightly to the DNA major groove with stable alpha-helix secondary structure

We hypothesized that downregulation of enzymes involved in DNA synthesis as a consequence of E2F inhibition, particularly TS and TK, would sensitize tumor cells to combinations of the D-Arg PEP with agents targeting these enzymes, e.g., pemetrexed, which targets TS (Figure 1). As E2F is also required for DNA repair [1,21], we also tested the combination with cisplatin, a DNA damaging agent. In this study, these combinations were tested *in*-*vitro* to treat prostate, breast, and lymphoma and with pemetrexed in NSCLC cells, and pemetrexed were used with PL-PEP in mice bearing the H2009 NSCLC tumor to determine *in*-*vivo* efficacy.

## 2. Results

### 2.1. D-Arg PEP in Combination with Cisplatin

Platinum compounds, e.g., cisplatin and carboplatin, have been used for the treatment of non-small cell lung cancer (NSCLC) and breast cancer in the clinic for decades. Cisplatin causes DNA damage and forms irreversible adducts with DNA, causing damage to replication and transcription processes [22]. E2F-1 functions to facilitate DNA repair by recruiting acetyltransferase tat interacting protein 60 (Tip60) complex subunits on E2F and targeting genes [23]. This Tip60 complex stabilizes E2F-1 by acetylation at lysine residues 120 and 125. Thus, the Tip60/E2F-1 complex controls the accumulation of the enzyme excision repair cross-complementing group 1 (ERCC1), known to play a rate limiting role in the repair of platinum-DNA adducts [24]. We hypothesized that the D-Arg PEP (as an inhibitor of E2F transcription) in combination with cisplatin would enhance DNA damage. The D-Arg PEP in combination with cisplatin showed additive or modest synergistic inhibition of the androgen sensitive DU145 prostate cell line and the castrate resistant PC3 prostate cell line (Figure 2 and Appendix A).

The D-Arg PEP in combination with cisplatin against breast cancer cell lines (MCF7, MDA-MB-468 and 4T1) and the lymphoma cell, CCRF-CEM also showed a modest synergistic anti-cancer effect (Figure 3A–D, Appendix A), when treated with D-Arg PEP and cisplatin. The γ-H2AX signal as a sensitive molecular marker for DNA damage after combination treatment is shown in Figure 4.

### 2.2. D-Arg PEP Showed a Marked Synergistic Anti-Cancer Effect with Pemetrexed against NSCLC Cell Lines

As treatment of cells with the PEP downregulated TS and TK due to inhibition of E2F-1 expression [19], it was of interest to determine if treatment of tumor cells with the combination of the D-Arg PEP and pemetrexed, a potent thymidylate synthase inhibitor [25,26], would result in additive or synergistic antitumor effect. As thymidine can rescue cells from TS inhibition, downregulation of TK by the PEP, thus preventing thymidine utilization, would add to the inhibition of TS and the anti-tumor effects of pemetrexed. As pemetrexed is used to treat NSCLC, we tested four adenocarcinoma lung cancer cell lines, H2009, H441, H1975, and H2228. The four cell lines showed different sensitivities to the PEP and pememtrexed (Appendix A). Fixed ratios of the D-Arg PEP to pemetrexed in the combination were chosen based on the 50% inhibitory concentration (IC_50_) of the cells to the individual compounds and combination indices to determine synergy, additivity, or antagonism [27]. Despite the differences in sensitivity to pemetrexed, the D-Arg PEP in combination with pemetrexed showed marked synergistic cell kill in all of the NSCLC cell lines—H1975, H2228, H2009, and H441 (Figure 5A–E). The Western blot analysis of the H2009 cell line after treatment with D-Arg PEP, pemetrexed, or combination showed that D-Arg PEP caused downregulation of E2F-1, TS, and TK, which was further enhanced when combined with pemetrexed (Figure 5F).

### 2.3. In-Vivo PL-PEP Combination Study with Pemetrexed

Based on these results showing synergistic cell kill of lung adenocarcinoma cells line with the combination of pemetrexed and the D-Arg PEP, we tested this combination in a mouse model. We selected the H2009 cell line for *in*-*vivo* study, as the H2009 cell line showed high levels of E2F-1, thymidylate synthase, and thymidine kinase expression. The binding of retinoblastoma protein (Rb) with the E2F-1 acts as a transcriptional repressor for the E2F targeting genes, and the cells arrest in G1 phase of the cell cycle. The loss of Rb by the deletion or mutation or phosphorylation of Rb by cyclins and cyclin-D kinases (cdks) leads to release of free E2F-1 which causes transcriptional activation of DNA synthesis genes, and thus the cells proceeds from G1 to S-phase of the cell cycle. Thus, phosphorylation of pRB due likely to CDK 4, 6, and 2 in three of the four cell lines resulted in high levels of E2F-1 as shown by Westerns (Figure 6). The level of TS and TK in the four cell lines showed no significant difference, so the difference in sensitivity to pemetrexed in the cell lines was not associated with an increase in TS protein (Figure 6).

The D-Arg PEP release profile was also checked from the PEGylated liposomes, and no substantial release was registered when PEGylated liposomal D-Arg PEP (PL-PEP) was incubated in saline (Appendix A). Less than 10% of the release was recorded in serum during 96 h of incubation. The data showed that PEGylated liposomes were very stable in serum and released the D-Arg PEP only after penetration to cancer cells and disruption inside the cells.

In the H2009 xenograft study, the combination of the D-Arg PEP and pemetrexed had an enhanced anti-tumor effect compared to either of the drugs (Figure 7A), without any obvious toxicity as shown by stable body weight throughout the experiment (Figure 7B). The PEGylated liposomal D-Arg PEP (PL-PEP) was also found to be effective against the H2009 cell line (*in*-*vitro*) with IC_50_ of 105 µM (Appendix A). The tumors harvested at the end of the experiment (Appendix A) were used for immunohistochemistry to test for various biomarkers. There was a significant decrease in Ki-67 staining in the combination group, showing that the PL-PEP and pemetrexed combination decreased proliferation of the NSCLC tumor. Furthermore, the TS and TK staining showed a significant decrease in the PL-PEP and combination group compared to the control and pemetrexed group (Figure 8). Thus, the penetratin-conjugated E2F peptide (inhibiting E2F-1, E2F-2, and E2F-3a) caused downregulation of TS and TK expression, which was further enhanced when combined with pemetrexed. The pemetrexed group not showing a decrease in TS and TK staining (compared to control group) could be that the enzyme levels recovered.

## 3. Discussion

It is well documented that many cancers, including some lung cancers, have deletions or mutations in the retinoblastoma protein that result in increased levels of “free” E2F-1 [18,19]. Of relevance, knockdown of E2F-1 was found to decrease the stem cell population in chronic myeloid leukemia without affecting the stem cell population in normal bone marrow [28]. This finding may explain the lack of toxicity we observed with the administration of the peptide.

The current research utilized an E2F peptide made more effective by modifications to decrease protease degradation, increase cellular uptake, and increase bioavailability. Substitution of L-amino acids in peptides by D-amino acid have been shown to make peptides more stable and increase their half-lives in serum [29,30] and could make D-amino acid peptides candidates for oral administration if they resist digestive enzyme degradation. This motivated us to modify the novel 7-mer peptide by substituting the L-Arg with D-Arg at the fourth position of the peptide. The alpha helix of D-Arg PEP is more stable energetically, and its binding to a major groove of DNA helix is more tight compared to L-Arg PEP [20]. *in*-*vitro* studies showed that the D-Arg PEP is more stable and more potent compared to L-Arg PEP when tested against the human castrate resistant cell line DU145 and the human lung cancer H196 cell line [20] and in the present study (Figure 8).

To increase the half-life of the PEP *in*-*vivo*, we encapsulated the peptide into PEGylated liposomes. In DU145 xenografts propagated in mice, PEGylated liposomal D-Arg PEP showed anti-tumor activity without toxicity [18,19].

To take advantage of the role of E2F-1 in DNA repair [1,21], we tested the combination of a DNA damaging agent, cisplatin in combination with the D-Arg PEP. The D-Arg PEP showed a modest synergistic effect in combination with cisplatin against prostate cancer, breast cancer and lymphoma cell lines, when analyzed using the Chou–Talalay method for synergy, additive, or antagonistic effects (Figure 2, Figure 3 and Figure 4). The D-Arg PEP, by inhibiting the activating E2Fs, resulted in downregulation of proteins required for DNA synthesis, in particular, TS and TK [19]. When tested in combination with pemetrexed, a potent TS inhibitor used to treat lung cancer patients, the combination caused marked synergistic cell kill in four adenocarcinoma cell lines. A xenograft study using the H2009 tumor showed anti-tumor effects of both the liposomal PEGylated D-Arg PEP and pemetrexed and enhanced effects of the combination, without any evident toxicity (Figure 5, Figure 7 and Figure 8).

## 4. Materials and Methods

### 4.1. Animals

Nude mice (CrTac: NCR-Foxn1<nu> both male and female) were obtained from the Taconic Biosciences, Inc. (New York, NY, USA).

### 4.2. Cell Lines and Chemical Compounds

Prostate cancer (DU145 and PC3), breast cancer (MCF7, MDA-MB-468, and 4T1), lymphoma (CCRF-CEM), and non-small cell lung cancer cell lines (H2009, H441, H1975 and H2228) were cultured in RPMI 1640 medium (Life technologies, Grand Island, NY) with 10% FBS (Thermofisher, NY, USA) and 1% penicillin/streptomycin (Thermofisher, NY, USA) in an atmosphere of 5% CO_2_. Cisplatin and pemetrexed were obtained from the Rutgers Cancer Institute of New Jersey (CINJ) pharmacy. The L- and D-Arg PEP were synthesized by Bio Basic Inc., Ontario, Canada, and validated by the Rutgers Chemistry Core as previously described [18]. MTS tetrazolium Promega CellTitre 96® Aqueous One Solution cell proliferation assay (G1111) was purchased from Promega, Madison, USA. Formaldehyde (18814) was purchased from Polysciences Inc., Warrington, USA. Triton™ X-100 (BP151), coverslips and glass slides were from Fisher Scientific, NY, USA, and Bovine Serum Albumin (A2153) was from Sigma-Aldrich, St. Louis, USA. All the cell lines were obtained from American Type Culture Collection (ATCC) and were checked for mycoplasma by MycoAlert™ mycoplasma detection kit (Lonza, Walkersville, MD, USA) before starting any experiment.

### 4.3. Cell Viability Assays

Five thousand cells were plated in a 96-well plate in 180 µL of respective media containing 10% FBS. After 24 h, 20 µL of either D-Arg PEP or cisplatin or pemetrexed (final dug concentration shown in results) was added and incubated for 48 h. Twenty µL of the MTS tetrazolium Promega CellTitre 96^®^ Aqueous One Solution cell proliferation assay was added to each well and incubated for 2 to 3 h. Absorbance was measured at 490 nm as per the manufacturer’s protocol, to determine the cell viability.

Five thousand cells (suspension) were plated in a 12-well plate in RPMI 1640 media supplemented with 10% FBS. After 24 h, the drug was added and incubated for 48 h. To assess cell viability, the cells with or without drug treatment were collected and cell viability was determined using the Vi-CELL^TM^ Series cell viability analyzer (Beckman Coulter, Carlsbad, CA, USA). The 50% inhibitory concentration (IC_50_; the drug concentration required to obtain 50% cell kill compared to control) was determined using the nonlinear regression curve fit of the graphs drawn by GraphPad Prism 4 software (GraphPad Software Inc., San Diego, CA, USA). All experiments were performed in triplicate, and all experiments were repeated at least three times.

### 4.4. In-Vitro Combination Drug Treatments

The cells were treated with either a single or combination of the drugs in a fixed ratio (based upon their IC_50s_). After 48 h after adding the drugs, cell viability was assessed by the MTS assay or Vi-CELL. For drug combination experiments, the cell viability assays were performed, and the results were analyzed for synergistic, additive, or antagonistic effects using the combination index (CI) method developed by Chou and Talalay by using Compusyn v1.0 software (ComboSyn, Inc., Paramus, NJ, USA). Combination indices, CI < 1, CI = 1, and CI > 1 indicate synergism, additive effects, and antagonism, respectively.

### 4.5. Immunofluorescence Assay

Cells were treated with IC_50_ concentration of either a single or combination of drugs for 48 h. Cells were then trypsinized, re-suspended in the fresh cell culture media, and were grown overnight on 20 mm coverslips in a 6-well plate (100,000 cells/coverslip). Next day, cells were washed with PBS, fixed with 4% paraformaldehyde for 15 min at room temperature (RT), washed three times with PBS, and permeabilized in 1% Triton X-100 at 37 °C for 30 min, followed by 10 min at RT and again washed three times with PBS. After blocking with 1% bovine serum albumin (BSA), washed cells were incubated with gamma H2AX (phospho-Ser139) monoclonal primary antibody (SAB5600038, Millipore Sigma, USA) at a dilution of 1:5000 overnight at 4 °C. The cells were incubated with goat anti-rabbit IgG-FITC (sc-2012, Santa Cruz Biotechnology, Dallas, TX, USA) at 1:2500 dilution for 2 h in the dark at room temperature and washed 3 times with the PBS. The coverslips were mounted on microscopic glass slides and sealed with the microscopic fluid on outer edges. The images were captured using a Nikon TE2000-U Fluorescence Microscope-AV (from the CINJ core facility) with 10× magnification.

### 4.6. Western Blotting

The expression of pRb, E2F-1, TS, and TK was checked by Western blot analysis using a standard protocol. Anti-E2F-1(sc-251), anti-TS(sc-33679), anti-TK(sc-56967), anti-vinculin (sc-73614), and secondary antibody were from Santa Cruz Biotechnologies, USA. Anti-pRb (9308S) was from the Cell Signaling Technology, Danvers, MA, USA.

### 4.7. Preparation and Characterization of Liposomes

Egg phosphatidylcholine (EPC), cholesterol, and 1,2,-distearoyl-sn-glycero-3-phosphoethanolamine-N-aminopolyethelenglycol—Mw—2000 ammonium salt (DSPE-PEG) were purchased from Avanti Polar Lipids, Alabaster, AL. PEGylated liposomes were prepared as previously described [31,32]. Briefly, lipids: EPC, cholesterol, and 1,2,-distearoyl-sn-glycero-3-phosphoethanolamine-N-aminopolyethelenglycol—Mw—2000 ammonium salt (DSPE-PEG) were dissolved in chloroform, evaporated to a thin film layer using rotary evaporator Rotavapor^®^ R-210/R-215 (BUCHI Corp., New Castle, DE, USA), and rehydrated with 0.9% NaCl to final lipid concentration 20 mM. The lipid mole ratio for this formulation was 51:44:5 EPC:Chol:DSPE-PEG, respectively. Liposomes were stored at room temperature for 1 h, followed by extrusion through polycarbonate membranes 200 nm and 100 nm using the extruder device (Northern Lipids Inc., Vancouver, BC, Canada). The size distribution and zeta potential of liposomes were measured using Malvern ZetaSizer NanoSeries (Malvern Instruments, Malvern, UK) according to the manufacturer’s recommendations. Measurements were performed at 25 °C. The sizes and zeta potential were measured five times, and average values were calculated. The average size of particles was 160 ± 10 nm; zeta potential was −10 ± 2 mV.

### 4.8. Loading Efficiency and Content of D-Arg PEP in the PEGylated Liposomes

D-Arg PEP was dissolved with lipids in chloroform, evaporated to a thin film layer using rotary evaporator Rotavapor^®^ R-210/R-215 (BUCHI Corp., New Castle, DE, USA), and rehydrated with 0.9% NaCl to final lipid concentration 20 mM. D-Arg PEP concentration was 10 mg/mL. In order to measure D-Arg PEP content in the liposomes, the aliquots of PEGylated liposomes containing D-Arg PEP were disrupted in isopropanol in a ratio of 10:90 (NLC/isopropanol), and the concentration D-Arg PEP was determined by high-performance liquid chromatography (HPLC) using a symmetry C18 column 150 mm × 4.6 mm (Water Corporation, Milford, MA, USA) operated at room temperature. The mobile phase consisted of 0.15 M NaCl, 20% acetonitrile solution; the flow rate was set to 0.4 mL/min, wavelength 280 nm. The chromatographic apparatus consisted of a Model 1525 pump (Waters Instruments, Milford, MA, USA), a Model 717 Plus auto-injector (Waters Instruments, Milford, MA, USA), and a Model 2487 variable wavelength UV detector (Waters Instruments, Milford, MA, USA) connected to the Millennium software. The average loading efficiency of liposomes was close to 95%.

### 4.9. D-Arg PEP Release Profile from PEGylated Liposomes

D-Arg PEP release experiments were performed in phosphate-buffered saline at pH 7.4 and temperature-equilibrated 25% human serum supplemented with RPMI nutrient media at 37 °C. The liposomes were put into dialysis tubing with a molecular weight cutoff of 10–12 kDa and dialyzed against 100 mL release buffers at 37 °C. At predetermined time intervals (0, 16, 32, 48, 64, 80, 96 h), samples were withdrawn from the release medium and assayed for D-Arg PEP by high-performance liquid chromatography as described above. Prior to final analysis, samples containing plasma were extracted by protein precipitation with 6% aqueous trichloroacetic acid, while samples without plasma were analyzed directly.

### 4.10. H2009 Xenograft Study

5 × 10^6^ H2009 cells with matrigel (1:1, *v*/*v* ratio) were injected subcutaneously in the abdominal right flanks of nude mice (CrTac:NCR-Foxn1<nu> both male and female). Once the tumor was palpable (around 2 weeks from cell injection), mice were randomized into four groups. One group of mice were injected with PL-PEP (80 mg/kg) every other day (3 times per week) and the other group with pemetrexed (250 mg/kg) twice a week. The control mice group received saline. The combination group received PL-PEP 80 mg/kg three times per week and pemetrexed 250 mg/kg twice a week. The mice were treated with either a single or combination of drugs for around 3 weeks. Tumor size and body weight were measured twice a week and tumor volume was calculated using (L)(W)^2^/2. Data were plotted and SEM was calculated.

### 4.11. Histologic Preparation and Immunohistochemistry Staining

Tumor samples extracted from the H2009 xenograft study at 7 days after last treatment were studied for immunohistochemistry. Samples were fixed in 4% formalin and paraffin embedded. Immunohistochemistry was performed on 4 µm sections with antibodies to Ki67 ((M3062) from Spring Bioscience Corp., Pleasanton, USA), TS (sc-33679), and TK (sc-56967) from Santa Cruz Biotechnologies, USA. Sections were developed and stained with hematoxylin and eosin using standard methods. All histological preparations and immunostaining were conducted by the Rutgers Cancer Institute of New Jersey Biospecimen Repository and Histopathology Core.

### 4.12. Statistical Analysis

All *in*-*vitro* experiments were performed three times, and each experiment was done in triplicate. Statistical analysis was performed using GraphPad Prism software. In all cases, ANOVA followed by two-tailed, unpaired Student’s *t*-tests were performed to analyze statistical differences between groups; *p*-values of < 0.05 were considered statistically significant.

## 5. Conclusions

The D-Arg PEP targeting activator family members of transcription factor E2F— namely, E2F-1, E2F-2, and E2F-3a—showed a synergistic anti-cancer effect when used in combination with chemotherapeutic drug cisplatin against the prostate cancer, breast cancer, and lymphoma cell lines. However, marked synergy was observed when used in combination with chemotherapeutic drug pemetrexed, a potent thymidylate synthase inhibitor, against the non-small cell lung cancer both *in*-*vitro* and *in*-*vivo*.

## Figures and Tables

**Figure 1 cancers-13-00972-f001:**
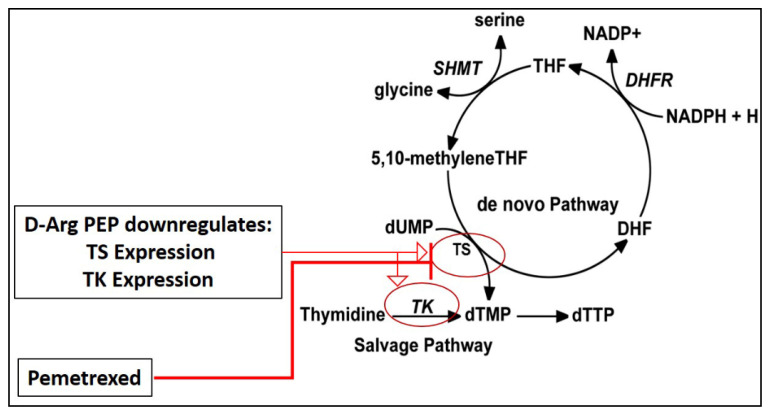
Schematic diagram showing D-Arg PEP and pemetrexed combination rationale. D-Arg PEP downregulates thymidylate synthase (TS) and thymidine kinase (TK) expression, and pemetrexed is a potent TS inhibitor.

**Figure 2 cancers-13-00972-f002:**
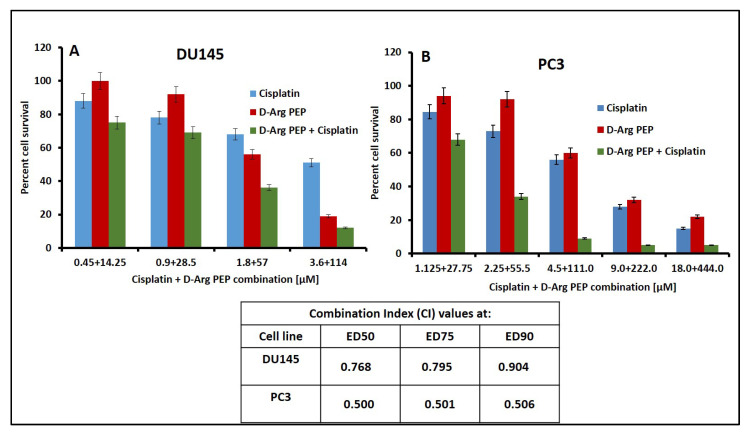
D-Arg PEP shows synergistic cell kill in combination with cisplatin in the prostate cancer cell lines DU145 (**A**) and PC-3 (**B**). Cisplatin and D-Arg PEP concentrations used for combination plates are shown on *x* axis and the percent cell survival (compared to untreated control) is shown on *y* axis. These drugs were added in combination at constant ratio as per 50% inhibitory concentration (IC_50_) value for individual drugs. The given concentration was also added as a single drug in individual plates. The combination index (CI) values obtained by Chou–Talalay analysis are tabulated below the figure.

**Figure 3 cancers-13-00972-f003:**
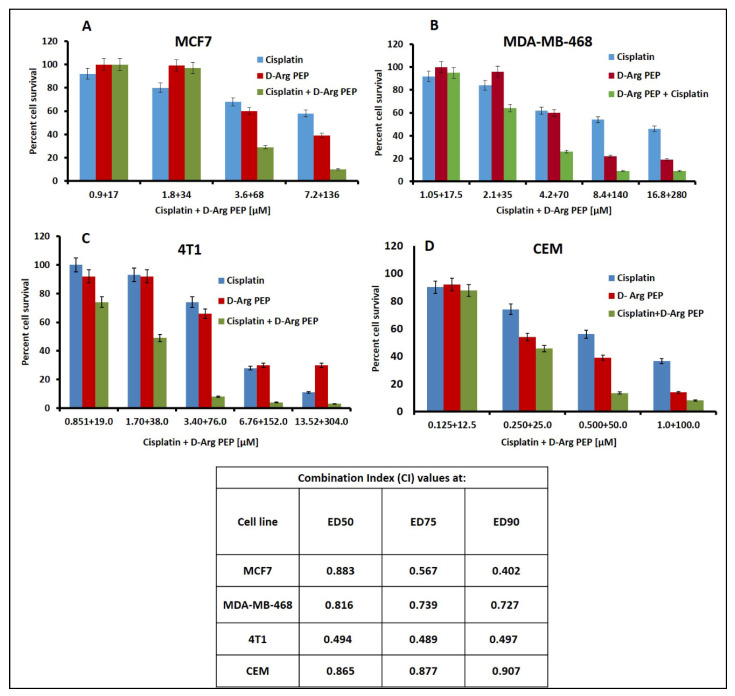
D- Arg PEP shows synergistic cell kill in combination with cisplatin in the breast cancer (**A**–**C**) and lymphoma (**D**) cell lines. Cisplatin and D-Arg PEP concentration used for combination plates are shown on *x* axis and the percent cell survival (compared to untreated control) is shown on *y* axis. These drugs were added in combination at constant ratio as per IC_50_ value concentration for individual drugs. The given concentration was also added as a single drug in individual plates. The combination index (CI) values obtained by Chou–Talalay analysis are tabulated below the figure.

**Figure 4 cancers-13-00972-f004:**
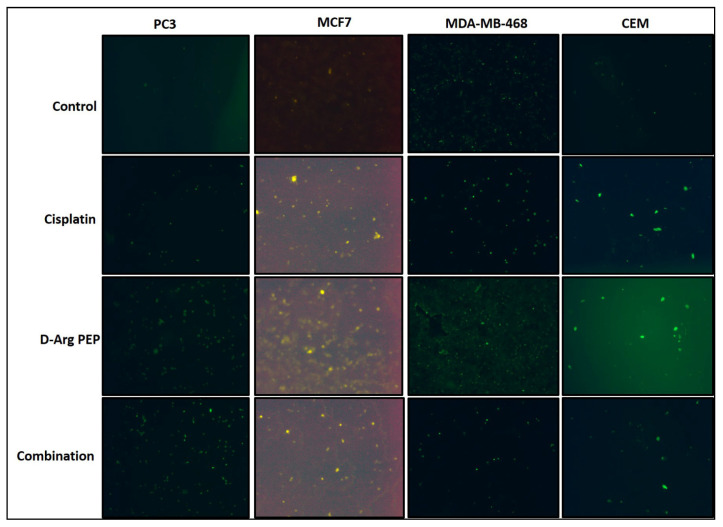
D-Arg PEP causes DNA damage. A synergistic effect in combination with cisplatin against the prostate cancer, breast cancer, and lymphoma cell lines is shown in Figure 2 and Figure 3. Immunofluorescence pictures of cells showing a FITC-γ-H2AX signal after 48 h treatment of cells with IC_50_ concentrations of D-Arg PEP or cisplatin or the combination compared to the untreated control. Nikon TE2000-U Fluorescence Microscope-AV (from the Rutgers Cancer Institute of New Jersey core facility) using 10× magnification was used to capture the images. It should be noted that the MCF7 and CEM cells showed a strong γ-H2AX signal that corresponds with their lower IC_50_ for D-Arg PEP.

**Figure 5 cancers-13-00972-f005:**
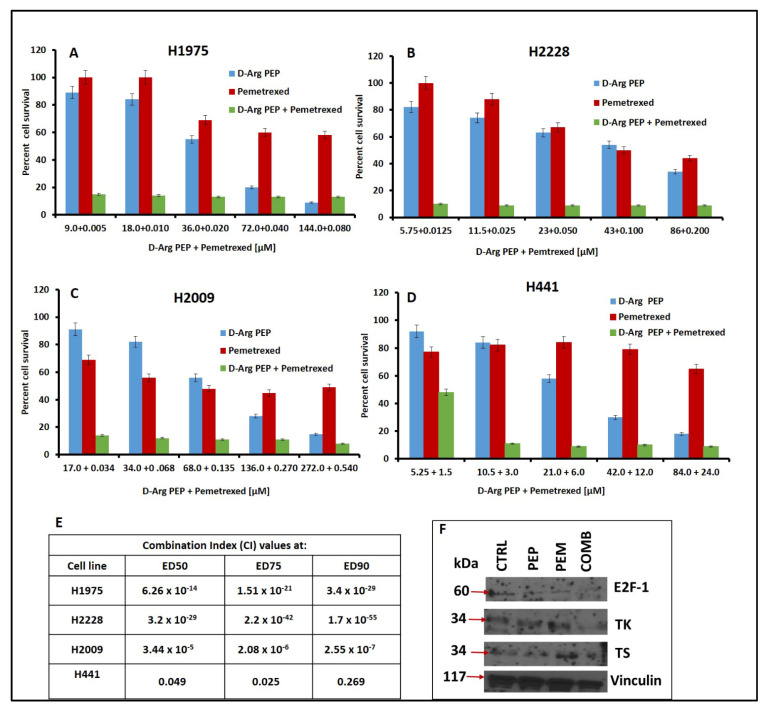
D-Arg PEP is highly synergistic in combination with pemetrexed against non-small cell lung cancer cell lines (**A**–**D**). D-Arg PEP and pemetrexed concentrations added in combination plates are shown on *x* axis and the percentage cell survival (compared to untreated control) is shown on *y* axis. The drugs were added in combination at constant ratio as per IC_50_ value concentration for individual drugs. The given concentration was also added as a single drug in individual plates. The combination index (CI) values obtained by Chou–Talalay analysis are tabulated (**E**). Western blot showing E2F-1, thymidine kinase (TK), thymidylate synthase (TS), and vinculin (loading control) in the H2009 cell line after treatment. Lane 1: untreated control; lane 2: D-Arg PEP (IC_50_ concentration); lane 3: pemetrexed (IC_50_ concentration); lane 4: combination of D-Arg PEP and pemetrexed (**F**). The uncropped Western blots are shown in Appendix A.

**Figure 6 cancers-13-00972-f006:**
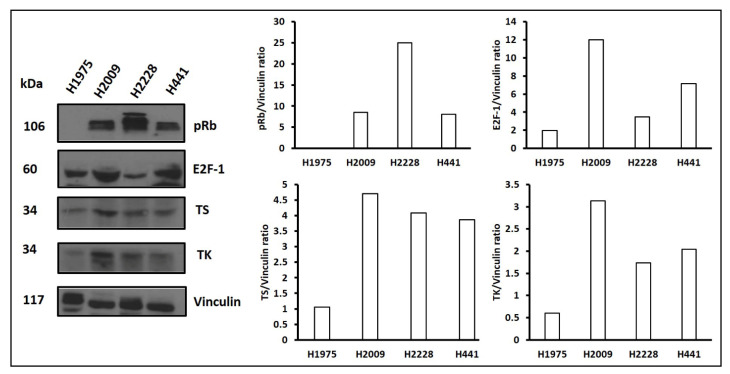
Western blot analysis of four non-small cell lung cancer cell lines. 12% SDS-PAGE was used and probe for phosphorylated retinoblastoma (pRb), E2F-1, thymidylate synthase (TS), thymidine kinase (TK), and vinculin as a control. The uncropped Western blots are shown in Appendix A.

**Figure 7 cancers-13-00972-f007:**
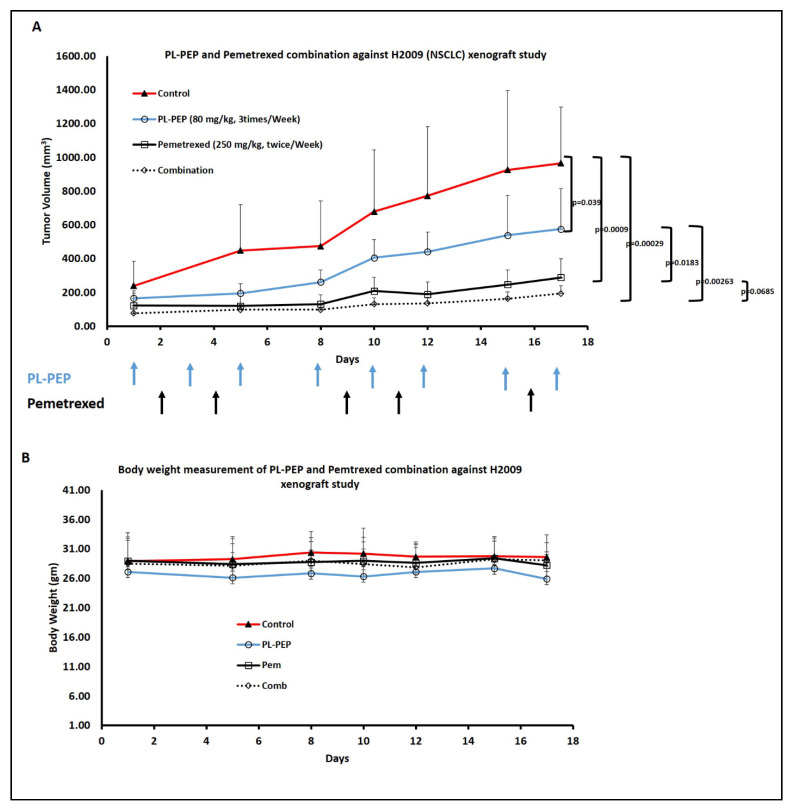
PEGylated Liposomal D-Arg PEP (PL-PEP) in combination with pemetrexed inhibited growth of H2009 NSCLC xenografts in mice. 5 × 10^6^ H2009 cells with matrigel (1:1 *v*/*v* ratio) were injected subcutaneously in right flank of nude mice. Once the tumor was palpable (around 2 weeks from cell injection), mice were randomized into 4 groups (*n* = 5). Mice were treated i.p. as follows: control group (saline, twice per week); PL-PEP group (80 mg/kg, 3 times per week); pemetrexed group (250 mg/kg, twice per week); and combination group (PL-PEP at 80mg/kg, 3 times per week and pemetrexed at 250 mg/kg, twice per week). The mice were treated with either a single or combination of drugs for around 3 weeks. Each drug treatment is shown by arrows. Tumor size and body weight were measured twice a week. Tumor volume (**A**) was measured using (L) (W)^2^/2. The combination of PL-PEP and pemetrexed treatment showed no change in body weight through the course of the experiment (**B**). Data were plotted and SEM was calculated. *p*-values < 0.05 were considered significant.

**Figure 8 cancers-13-00972-f008:**
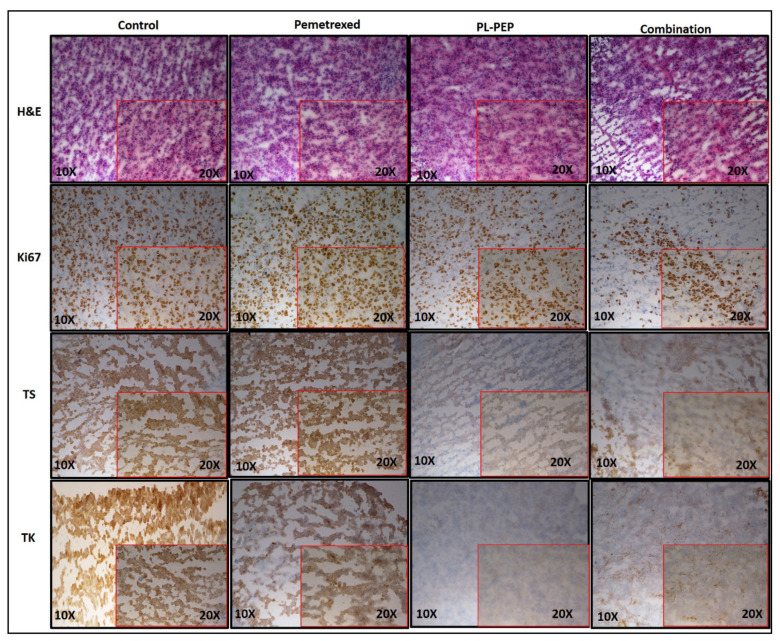
Immunohistochemistry (IHC) staining of H2009 tumors extracted from mice 7 days following the last dose of drug treatment in Figure 7 against various biomarkers. Ki67 staining showing proliferation; TS (thymidylate synthase) and TK (thymidine kKinase) are key enzymes involved in DNA synthesis. Tonsil tissue was used as a positive control for IHC staining.

## Data Availability

All the data in this study is contained within the article or Appendix A.

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
