# Peer review of "Anti-Tumor Effects of a Penetratin Peptide Targeting Transcription of E2F-1, 2 and 3a Is Enhanced When Used in Combination with Pemetrexed or Cisplatin"

_cancers, 2021, doi:10.3390/cancers13050972_

Round 1

Reviewer 1 Report

Rather and coauthors investigate the synergy effect of a penetratin peptide and small chemotherapy drugs (pemetrexed or cisplatin). The results are interesting. However, it could be suggested for publication in Cancers once the following issues are addressed.

  1. Please add the loading efficiency and content of drug-loaded PEGylated liposomes.
  2. The drug release profile for drug-loaded PEGylated liposomes in vitro should be provided.
  3. All the in vitro cell experiments didin't check drug-loaded PEGylated liposomes, which should be added.
  4. Line 229, ‘PEGlyated’; Line 281, ‘IC50s’; Line 323, ‘25ºC’.

Reviewer 2 Report

The revised manuscript is improved. 

Reviewer 3 Report

Dear Editor,

I would like to ‘Thank’ the authors for acting positively on the comments raised for the initial version of this manuscript.  I am very pleased with the new version and out of the 6 points raised by myself, 5 appear to have been addressed with satisfactory outcomes.  I am however, sorry to say that one important experiment is outstanding (see point number 3-, below) and which is a point that has not been fully addressed by the authors.  As stated, this is a critical piece of data that links the cell line model to the xenograft model and highlights a clearer link between the treatments used in the in vitro studies, in vivo studies and gene regulatory events as a key mechanism. It would be helpful if the authors could address this point experimentally.

1-Response: Thank you for your comments.

Yes, as mentioned in our MS, previously we showed that PEP (Penetratin E2F

Peptide) downregulates thymidylate synthase (TS) and thymidine kinase (TK)

expression (ref. 18 and 19). Thus, sensitizing these tumors to pemetrexed, a

potent TS inhibitor, showed marked synergistic effect in combination with PEP in

lung adenocarcinoma both in-vitro and in-vivo.

The functional roles of TS and TK, are the two important enzymes required for

DNA synthesis via de-novo and salvage pathway, respectively (see Figure 1 in the

revised MS).

Thank you for including the diagram for figure 1-it really adds a lot to the manuscript and helps the reader enormously.

2-Response: We showed that PEP inhibits expression of E2F1, TS and TK-

(References 18, 19).

In reference 18, H-69 cells are used Ref 18. H69 cells used-PEP IC50 is 7.9, which is very different to the IC50 observed with cell lines treated with D-Arg alone in figures S3 and S4 (of this manuscript), highlighting fundamental biological differences between the cell used in this study and the ones referenced. In reference 19, prostate cancer cells are used (Bu145, PC3, LnCaP). As stated in my original comments, if a general mechanism is to be defined then experimental approaches/treatment conditions used in Refs 18 and 19 have to be used for cell lines H2009 etc (from this manuscript), to show that the cells are behaving in a manner that is similar to other cells lines.

3-As mentioned in MS, the purpose of figure 6 (western

blotting-earlier figure 4) was to check the expression of E2F1 (as a response to

PEP), TS (response to pemetrexed) and TK (expression downregulated with the

PEP treatment) among the 4 NSCLC cell lines in order to select one for the in-vivo

study as all the 4 cell lines showed a marked synergistic effect (Figure 5-

combination indices).

Old Figure 4 shows protein expression levels the absence of treatments. Old Figure 6 is the xenograft model with treatments. My point (that appears to have been missed) was that surely to reconcile xenograft data and implicate TS and TK regulation as a mechanism, the stated genes have to be shown to be regulated in the cell lines being used in the xenograft model (under stimulatory conditions used in the old version of Figure 6 (the new Figure 8). In light of these suggestions, the old Figure 4 and legend needs to be extended.  This is a critical (and necessary) piece of data that defines a causal link between how the cell lines behave in response to treatments (in vitro) in relation to the effects observed with rescued tumors from the xenograft studies (as clearly shown in Figure 8).

4-Also, as shown (figure 6) the H2009 cell line showed higher expression of E2F1,

TS and TK compared to other 3 cell lines including H441; the reason for selecting

it for the in-vivo xenograft study.

This is clear but the westerns have to be shown for these genes after treatments (used for Figure 8) to implicate gene regulation as being affected by peptide/Pemetrexed treatment combinations (as suggested above in Point 3-).

5-The reason for not selecting the H441 cell line

for the in-vivo study was that it was known to be a pemetrexed resistant cell line

(https://utswmed-ir.tdl.org/handle/2152.5/1101 -see page 18), and our data also

showed that only 30% cell kill at 24 μM concentration of pemetrexed in the H441

cell line compared with H2009.

Thank you, but H2009 (from Figure 5) doesn’t appear to be hugely sensitive to the effects of pemetrexed either.

6-Response: We investigated our hypothesis further by performing an

immunohistochemistry assay with the H2009 tumors, from the mouse study to

check whether there is an effect on TS and TK as shown previously in the prostate cancer cell line DU145 (ref.18). We updated these results in the MS (line 185-193) as that the tumors harvested at the end of the animal experiment (Figure S5) wereused for immunohistochemistry to test for various biomarkers. There is a

significant decrease in Ki-67 staining in the combination group, that showed that

the PL-PEP and pemetrexed combination decreased proliferation of the NSCLC

H2009 tumor. Furthermore, TS and TK staining showed a significant decrease in

the PL-PEP and combination group compared to the control and pemetrexed group (Figure 8), thus highlighting that the penetratin-conjugated E2F peptide (by

inhibiting E2F-1, E2F-2, and E2F-3a) caused downregulation of TS and TK

expression which is further enhanced when combined with pemetrexed.

Thank you. This is a very good figure and supports the Xenograft data from the original MS version very nicely.

Round 2

Reviewer 1 Report

I recommend it for publication in the current form.

Reviewer 3 Report

Dear Authors,

Thank you for adding Figure 5F to the manuscript, in addition to fully addressing the points raised.  In balance, the addition of Fig5F certainly makes the paper 'more complete' and offers greater support to the conclusions being drawn. Well done.

This manuscript is a resubmission of an earlier submission. The following is a list of the peer review reports and author responses from that submission.

Round 1

Reviewer 1 Report

Anti-tumor Effects of a Penetratin Peptide Targeting Transcrip-2 tion of E2F-1, 2 and 3a is Enhanced When Used in Combination 3 with Pemetrexed or Cisplatin

Gulam M. Rather et al

The authors address the potency of D-Arg PEP in combined treatments with Cisplatin or Pemetrexed in reducing cell survival of a panel of cell lines using MTT assays.  Such findings are also extended to define an IC50 for each treatment using BC, PC and NSLC cells. From the panel of NSLC lines, they choose to utilize H2009 in a Xenograft model to address the efficacy of the combined treatments in reducing tumor size.  Under the conditions utilized, a significant reduction in tumor volume was observed highlighting the potential of D-Arg PEP in sensitizing tumors to the cytotoxic effects of Pemetrexed.

The hypothesis being tested is whether inhibition of E2F down regulates TS or TK, thus predisposing them to be sensitized to Pemetrexed, as the underlying mechanism

Minor considerations:

Generally speaking, the figures are well laid out and clear to follow, as is the text (although it is a little confusing to follow in places.  The essence of the paper is somewhat descriptive and additional details highlighting mechanistic details.  For example, what are the functional/biological roles of TS and TKs from mouse knockout studies etc? This would offer a heightened degree of physiological relevance as mouse models are utilized within the study.

Major Considerations:

-While the manuscript is descriptive in terms of the physiological effects of the combined therapeutics, I think what is lacking is some biochemical or mechanistic insight to either support of reject the major hypothesis being tested.  For example, the only western shown is in Figure 4 to highlight the rationale for using H2009.  I think what is required is another set of westerns to show how E2F1, TS, TK or Vinculin may be modulated based on the treatments used in the mouse experiments (shown in Figure 5) for some of the cell lines, including H2009.  Moreover, H441 has a similar profile to H2009 for the aforementioned proteins and I think the authors need to elaborate on why H441 wasn’t used as a way of defining how well the hypothesis holds up. 

-The findings (from the cell lines and the mouse work) are not particularly well discussed in the Discussion section in relation to the hypothesis being tested. Moreover, within the Discussion, it is stated that D-arg-PEP has a negative effect on E2F1, E2F2, E2F3, TS and TK protein levels-and which it does, albeit in Prostate Cancer Du145 cells (ref 19).  I think to extend such an observation into an alternative type of cancer or cell line system (eg NSCLC cells, Figure 5), there has to be a clear demonstration that the biochemical effects are similar to the effects observed in prostate cells, in order to establish consistency and a possible general mechanism (which is what the authors are generally proposing).   

Reviewer 2 Report

Manuscript ID: cancers-1067607

Title: Anti-tumor Effects of a Penetratin Peptide Targeting Transcription of E2F-1, 2 and 3a is Enhanced When Used in Combination with Pemetrexed or Cisplatin

Authors: Gulam Mohmad Rather, Michael Anyanwu, Zoltan Szekely, Joseph R. Bertino *

Submitted to section: Cancer Therapy

Manuscript submitted by Rather et al., describes the antitumor effects of a modified E2F peptide substituting D-Arg for L-Arg, conjugated to penetratin (PEP) against solid tumour cell lines and the CCRF-leukemia cell line, alone and in combination with pemetrexed or with cisplatin. Authors demonstrated that he D-Arg PEP in combination with cisplatin caused synergistic cell kill against prostate, breast, lung cancers and the CCRF-CEM cell line. Marked synergy resulted when the D-Arg PEP was used in combination with pemetrexed against the lung adenocarcinoma cell lines. A xenograft study using the PL-PEP in combination with pemetrexed showed enhanced anti-tumour effects compared to each drug alone.

My opinion is that the manuscript in this form is not sufficient to be accepted by Cancers. There are several reasons for this opinion:

  • I general, the manuscript lack information in regard to signalling pathway which is responsible for synergistic effect of PEP and cisplatin (cDDP). If authors hypothesis that PEP in combination with cDDP probably enhance DNA damage they need to show it. More detail analysis of this specific effect is needed. There are so many data how cDDP works and this specific drug is broadly used. Authors need to show how PEP really adds value in cDDP toxicity increase.
  • In addition, they need to show that PEP alone and in combination with cDDP is less toxic to healthy cells.
  • Figures 1 and 2 are very confusing. Titles of X and Y axis need to be more precise and the legend for each data presentation is needed! It is really not clear what dose numbers indicated on axis X mean. Moreover, both figures are prepared very clumsy since sometimes the name of Y axis is completely missed and sometimes only half described (lacking μM for example). Both tables in both figures are also very clumsy prepared and it is not clear which concentrations of compounds are used actually. Figures descriptions are misleading. The material and methods part of the manuscript is a part where methods are described…not figure description where details shown in figures need to be indicated. All figure descriptions need to be rewritten (concentrations used, meaning of shortcuts etc.).
  • Authors need to decide will they use IC50 or IC50 etc., E2F1 or E2F-1 for example. Then, in vivo (vitro) in italic I usually used…not in-vivo (vitro).
  • In general, name of each figure need to be rewritten in a same way for each figure! Otherwise, it is very confusing. I am recommending that each figure has a title which is actually the conclusion message of each set of data shown in the figure.
  • In Figure 4…why NSCLS cell lines is written four times? Then complete name of the protein (the same is already written in figure description) and etc…
  • Figure 6 needs mice/tumours photos!

Reviewer 3 Report

The authors address a highly interesting and needed study concerning to the synergistic anti-cancer effect of a penetratin peptide in combination with pemetrexed or cisplatin. Despite the interest of the subject, the manuscript lacks several aspects and should be deeply revised before its consideration for publication in Cancers.

Major points

  1. Figure 3, for the H1975 cell line, there is no synergistic effect at high concentration. Please make some discussion about this point.
  2. All the in vitro cell experiments were done with PEGylated liposomes, such experiment should be done at least with H2009 using PEGylated liposomes.
  3. All the information about the chemicals used should be added in the section 4.2.
  4. There is a lack of basic information for a research publication, such as the size and zeta potential of liposomes w/o penetratin, pemetrexed or cisplatin.
  5. There is also a lack of the preparation detail of liposomes with penetratin, pemetrexed or cisplatin. With the carriers, there is huge change of the drug efficiency. I don’t think the in vitro and in vivo experiment are so relevant.

Minor points

  1. Figure 1-3 are not sharp enough.
  2. Figure 1: there is a lack of the meaning of the axis.
  3. There are references related to the subject more recent that the referred by the authors.
  4. Format issues: for example, half -life (line 23), D- Arg (line 111), IC50 (line 113), CO2 (line 227), (L)(W)^2/2 (line 273). There are much more format issues all through the ms., please revise it carefully.
